# Layer Composition of Continuously Reinforced Concrete Pavement Optimized Using a Regression Analysis Method

**Byoung Hooi Cho** [1] , **Moon Won** [2] **and Boo Hyun Nam** [3],*

1 Department of Civil Engineering, Sangmyung University, Cheonan 31066, Korea; byoungcho@smu.ac.kr
2 Department of Civil, Environmental, and Construction Engineering, Texas Tech University, Lubbock, TX 79409, USA; moon.won@ttu.edu
3 Department of Civil, Environmental and Construction Engineering, University of Central Florida, Orlando, FL 32816, USA
* Correspondence: boohyun.nam@ucf.edu

**Abstract:** A procedure for determining the optimized composition of layer properties for a continuously reinforced concrete pavement (CRCP) system was constructed using field tests, finite element (FE) analysis, and regression analysis methods. The field support characteristics of a rigid pavement system were investigated using a falling weight deflectometer (FWD), dynamic cone penetrometer (DCP), and a static plate load test. The subgrade layer exhibited a more uniform condition than the aggregate base, and the modulus of the subgrade reaction of the aggregate base and subgrade combination (effective $k$-value) was improved by about 1.5 times by introducing a 2 inch (50.8 mm) asphalt stabilized base (ASB) layer. Thereafter, FE support models describing the actual field conditions were studied. The effects of the thickness of the stabilized base layer, the elastic modulus of the stabilized base material, and the effective $k$-value on the composite $k$-value of the support system were identified using a regression analysis method, and the results showed that the variables had a similar effect when determining the composite $k$-value. Afterward, a procedure for selecting the layer properties for producing a suitable composite $k$-value was constructed, and we identified that the maximum stress in the concrete slab was induced at different levels, even with identical composite $k$-values. Lastly, regression relationships were derived to estimate the maximum stress in the concrete slab by considering both the support layer properties and the concrete slab. Subsequently, an algorithm for selecting an optimized layer composition of the CRCP structure was construction considering the economical aspect.

**Keywords:** continuously reinforced concrete pavement (CRCP); finite element support model; regression analysis method; stabilized base layer; effective $k$-value; composite $k$-value; optimized layer composition

## 1. Introduction

Continuously reinforced concrete pavement (CRCP) is the most widely constructed type of Portland cement concrete (PCC) pavement in Texas. After the first CRCP construction in Columbia Pike near Washington, DC, USA, in 1921, several states, including Texas, Virginia, Louisiana, Georgia, North and South Dakota, Oklahoma, Indiana, Illinois, California, and New Jersey, constructed CRCP [1]. In particular, Texas has completed more CRCP construction projects than any other state in the United States due to the state's policy, which states that newly established concrete highways must be constructed from CRCP unless there are specific reasons to do otherwise [2]. It is well-known that the CRCP system has no artificial transverse joints and allows for irregular transverse cracks because the embedded longitudinal reinforcement holds the cracks tightly, resulting in the CRCP showing excellent performance for heavy vehicle loads. Therefore, a well-designed and constructed CRCP system can be expected to provide adequate performance with minimal maintenance for over 40 years [3].

Previous studies reported that a support system of concrete pavement structures is one of the most critical factors for guaranteeing reliable rigid pavement performance [4–7]. It is clear that a strong and uniform subgrade layer condition can provide stable support for a concrete slab, whereas weak and nonuniform subgrade results in differing settlements, cause the pavement system to become damaged easily. Previous studies in the state of Virginia reported that a poor subgrade requires thicker CRCP slabs as compensation [8]. Illinois reported that a well-prepared subgrade for CRCP provides a smooth, low-maintenance ride for many years' worth of heavy traffic [9]. For these reasons, in general, one or more base layers are placed between the compacted subgrade soil and the concrete slab to minimize the damage and failures of a rigid pavement system by providing stable and uniform conditions. An adequate support system, in terms of both structural and functional aspects, especially a non-erodible base layer, can provide not only a stable construction platform and uniform slab support conditions, but also prevent erosion of the base and subgrade materials. Erosion and loss of support materials along a pavement shoulder and longitudinal joint have been revealed as critical factors for the development of punchout, which is a major type of distress in CRCP systems [4,10,11].

The CRCP support system generally consists of compacted subgrade soil and a stabilized base layers beneath a concrete slab. The modulus of the subgrade reaction (*k*-value) is one of the most important values for rigid pavement design. The *k*-value represents the relationship between an applied pressure load and the corresponding deflection of the top surface of a compacted subgrade soil layer. A group at The University of Texas at Austin has attempted to characterize the subgrade resilient modulus using FWD and correlate to the modulus of subgrade reaction [12,13].

Although the currently developed pavement design guides consider the *k*-value using a unique support characteristic, the *k*-value is not only affected by the size of the loading area, but also the thickness and modulus of the concrete slab [14,15]. However, these characteristics of the *k*-value are not incorporated into most of the current design algorithms. In some cases, an aggregate base layer is constructed on the subgrade layer, and the stiffness characteristic (a relationship between the pressure load and the deflection) on the top surface of the aggregate base layer, which includes the effect of the subgrade, is the effective *k*-value. In modern pavement design guides, a stabilized base layer, such as an asphalt stabilized base (ASB), a cement-treated base (CTB), or a lean concrete base (LCB), is highly recommended beneath a concrete slab [16]. In this case, the stiffness of the support structure on top of the stabilized base layer is called the composite *k*-value and is used as a crucial design factor.

Although the use of a stabilized base layer can provide an adequate construction platform and prevent the loss of material from the support layers, resulting in the good performance of a rigid pavement system, its construction costs are high. In Texas especially, the use of non-erodible stabilized base layers is required to prevent failures of the CRCP that are related to pumping and erosion of the support materials, resulting in a high initial construction cost. For this reason, it is highly desired to decrease the initial construction cost and provide acceptable long-term performance of the pavement system in terms of the structural and functional aspects.

In this study, we focused on developing a design algorithm for CRCP systems by determining an optimized composition of a layered CRCP system that can minimize the initial construction cost and provide acceptable performance during a designated lifetime. Initially, the field support conditions and characteristics of the rigid pavement structure were investigated using field tests: falling weight deflectometer (FWD), dynamic cone penetrometer (DCP), and non-repetitive static plate load (*k*-value) tests. Thereafter, using finite element (FE) and regression analysis methods, the effects of the design properties of the CRCP structure on the composite *k*-value and maximum stress induced in the concrete slab were discussed. Subsequently, a procedure for determining the optimized layer composition of the CRCP was constructed that considered both the structural capability of the support system and the economical aspect.

## 2. Field Investigation

Three types of field tests, including FWD, DCP, and non-repetitive static plate load tests, were conducted in accordance with American Society for Testing and Materials (ASTM) E2583 [17], ASTM D6951 [18], and ASTM D1195 [19], respectively, to investigate the field support structure conditions for rigid pavement at a test section located on the J.J. Pickle Research Campus of the University of Texas in Austin. Although the standard method of the static plate load test recommends the use of 24 in (610 mm) diameter loading plate, a 12 in (305 mm) loading plate was used in this study since it is hard to handle and requires a heavy reaction force. Through these field tests, the elastic modulus of each layer, effective and composite *k*-values, and uniformity of the support structure were investigated. The test site was composed of three layers: 2 in (50.8 mm) ASB layer, 8 in (203 mm) aggregate base layer, and compacted subgrade soil layer.

The FWD tests were conducted on the top surface of ASB layer, as shown in Figure 1a. The objectives of the FWD tests were to (1) check the uniformity of the support structure, and (2) obtain the elastic modulus of each layer. Figure 1b shows the deflection contour of the FWD test corresponding to a normalized dropping loading of 1000 lbs (4.45 kN), where the distance of each grid was 3 ft (914 mm) in both the x and y coordinates. As illustrated in Figure 1b, the field support structure showed irregular and non-uniform conditions. Therefore, the measured data were back-calculated to determine an elastic modulus of each layer, which were averaged, as shown in Table 1.

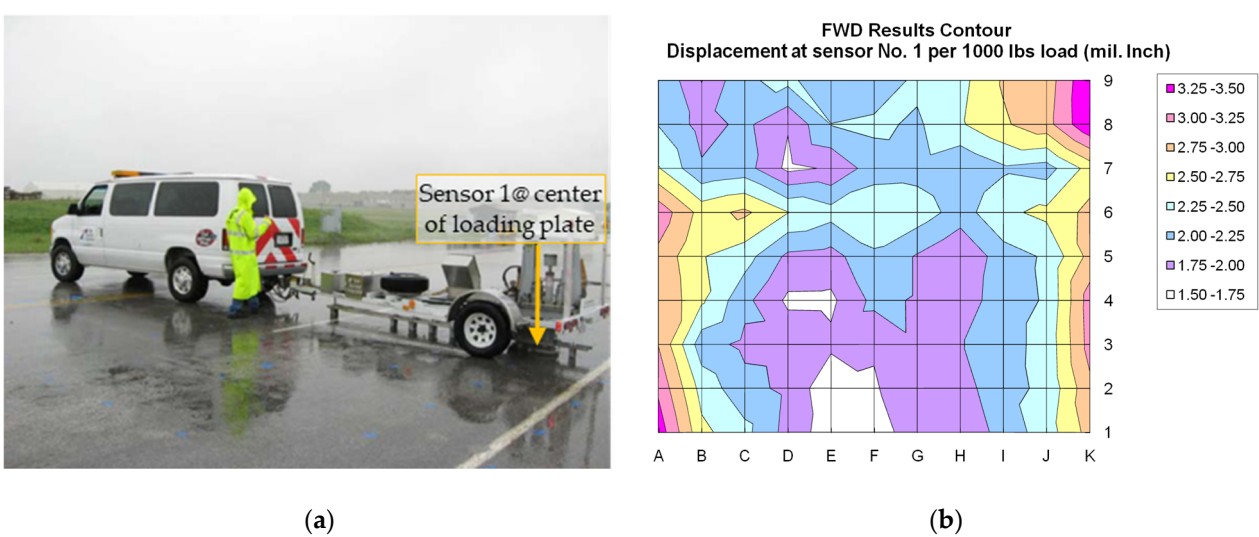

(**a**)                                                                 (**b**)

**Figure 1.** Falling weight deflectometer (FWD) test: (**a**) field overview and (**b**) deflection contour (1 lb = 4.45 N, 1 mil = $2.54 \times 10^{-2}$ mm) [20].

**Table 1.** Averaged back-calculated elastic modulus of each layer. (1 psi = 6.89 kPa).

| Layer | Average Elastic Modulus (psi) | Standard Deviation (psi) | Coefficient of Variance (%) |
|---|---|---|---|
| ASB * | 485,000 | 0 | 0.0 |
| Aggregate base | 38,400 | 9500 | 24.6 |
| Subgrade | 27,100 | 2900 | 10.7 |

*, ASB, asphalt stabilized base; elastic modulus of ASB material from a laboratory test was used as a fixed value.

Based on the FWD result, three locations were selected for DCP and non-repetitive static plate load tests, which are K8, K2, and I3 in Figure 1b, representing weak, medium, and strong support conditions, respectively. Since the DCP device cannot penetrate the ASB layer, holes were drilled in the three locations and the DCP tests were conducted from the top surface of the aggregate base layer into the subgrade soil. Figure 2 presents the results of

the DCP test. The boundary between the aggregate base layer and the subgrade soil, which was located at a depth of 8 inches, is clearly shown by presenting the changing trend lines. The results show that the subgrade soil showed more uniform condition than the aggregate base layer when considering the slopes of the graphs. The DCP data are expressed as DCP index (*DCPI*) described by penetration depth per blow (mm/blow). In this study, the correlation between *DCPI* and *CBR* (California Bearing Ratio) proposed by The U.S. Army Corps of Engineers (Equation (1)) [21] and another correlation between *CBR* and the resilient modulus (*M_r*) of the layers proposed by the Mechanistic-Empirical Pavement Design Guide (MEPDG) (Equation (2)) [22] were used to estimate layer properties.

$$CBR = \frac{292}{DCPI^{1.12}} \quad (\text{if } CBR > 10) \tag{1}$$

$$M_r(psi) = 2555(CBR)^{0.64} \tag{2}$$

Table 2 summarizes the CBR and $M_r$ values of the aggregate base and subgrade layers back-calculated from DCP results at the three locations marked as weak, medium, and strong by the FWD results. Although both DCP and FWD showed similar results qualitatively, the DCP produced somewhat larger $M_r$ values for the aggregate base layer (36,904–67,226 psi (254–464 MPa)) whereas the DCP estimated smaller values for the subgrade soil (14,181–16,309 psi (97.8–112.4 MPa)) than the FWD (38,400 psi (265 MPa) for the aggregate base and 27,100 psi (186.8 MPa) for the subgrade).

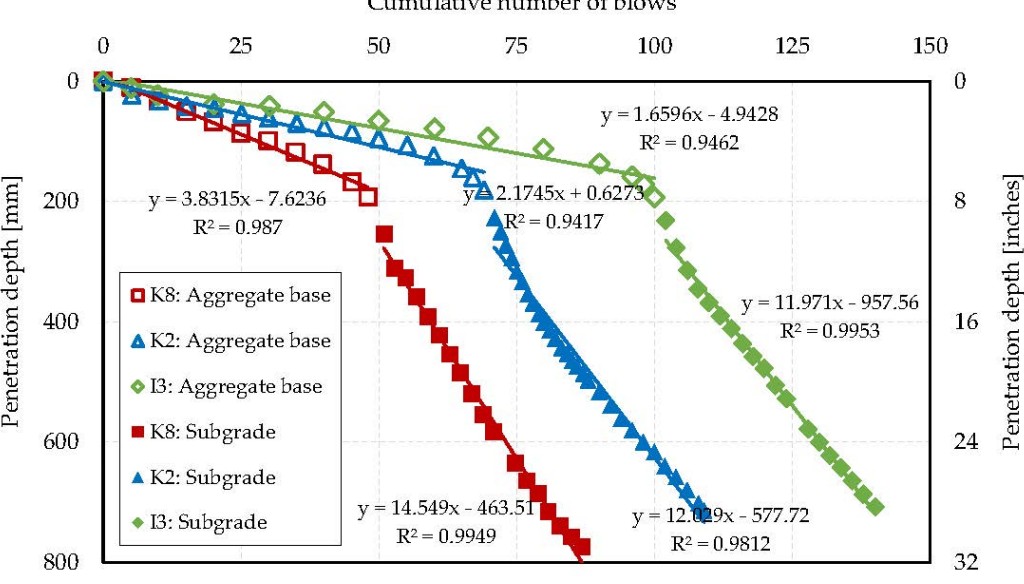

**Figure 2.** The results of the dynamic cone penetrometer (DCP) tests for the aggregate base and subgrade layers.

**Table 2.** Back-calculated properties of support layers by DCP results. (1 psi = 6.89 kPa). DCP index (DCPI), DCP index.

| Location | Layer | DCPI (mm/blow) | CBR | $M_r$ (psi) | Remark of FWD Result |
|---|---|---|---|---|---|
| K8 | Aggregate base | 3.8315 | 64.9 | 36,904 | Weak |
|  | Subgrade | 14.549 | 14.6 | 14,181 |  |
| K2 | Aggregate base | 2.1745 | 122.3 | 55,388 | Medium |
|  | Subgrade | 12.029 | 18.0 | 16,253 |  |
| I3 | Aggregate base | 1.6596 | 165.6 | 67,226 | Strong |
|  | Subgrade | 11.971 | 18.1 | 16,309 |  |

Non-repetitive static plate load tests were also performed on the surface of ASB layer at the K8, K2, and I3 location to investigate the composite *k*-values. In this field test, a steel load-bearing plate 12 in (305 mm) in diameter was used. A gross weight of reaction force of 48,000 lbs (213.5 kN) was prepared, and a vertical deflection of the ground surface was measured using two linear variable differential transformers (LVDTs) and a dial gauge. Once all devices were set up, the pressure load was increased until 0.005 in (0.127 mm) deflection was reached, and we waited until the rate of deflection increase stabilized at no more than 0.001 in/min ($2.54 \times 10^{-2}$ mm/min). This process was continued until the total surface deflection reached 0.05 in (1.27 mm). The composite *k*-values were then calculated at the 0.05 in (1.27 mm) deflection point. To investigate the effect of ASB on the composite *k*-value, this test was also conducted on the top surface of the aggregate base layer at location K2 after removal of the 2 in (50.8 mm) of ASB. Figure 3a,b show the test results and overview of field set-up, respectively. According to previous studies, a 12 in (305 mm) diameter loading plate tends to produce almost twice greater *k*-value than the use of a 30 in (762 mm) diameter plate, which is the standard [23–25]. For this reason, the obtained *k*-values from the field tests were corrected by dividing by two. Table 3 presents the composite *k*-values obtained from the field test and a comparison with classification per the FWD results. In conclusion, the test section shows the composite *k*-values ranging from 600 to 840 psi/in (162.9 to 228.0 MN/m$^3$) on the top surface of the ASB layer, and the trends in results well-match the FWD and DCP results. We also found that the ASB layer dramatically increased the stiffness of support system. In this study, the *k*-value increased from 420 to 680 psi/in (114.0 to 184.6 MN/m$^3$) due to the 2 in (50.8 mm) ASB layer.

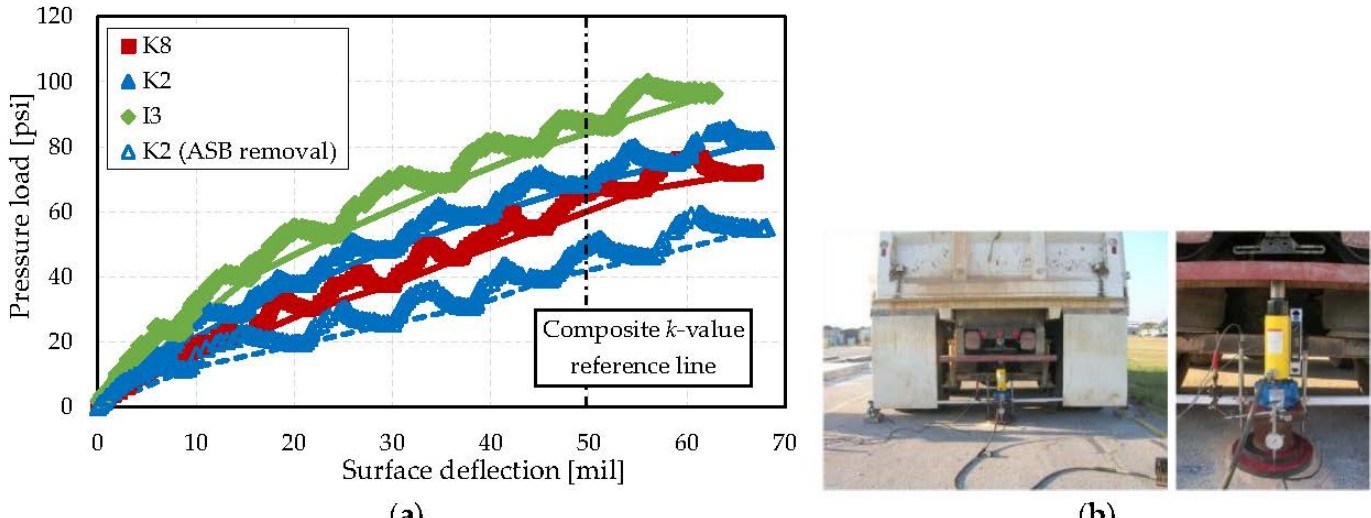

**Figure 3.** (**a**) Results of the non-repetitive plate load test, and (**b**) overview of the field set-up (1 psi = 6.89 kPa, 1 mil = $2.54 \times 10^{-2}$ mm).

**Table 3.** Composite *k*-value from non-repetitive static plate load test (1 psi/in = 271 kN/m$^3$).

| Location | Composite *k*-Value (psi/in) | | Remark of FWD Result |
| --- | --- | --- | --- |
| | **Field Test (12 in Loading Plate)** | **Corrected (30 in Loading Plate)** | |
| K8 | 1200 | 600 | Weak |
| K2 | 1360 | 680 | Medium |
| I3 | 1680 | 840 | Strong |
| K2 (ASB Removal) | 840 | 420 | - |

### 3. Evaluation of Support Models

To select the appropriate support model, the non-repetitive static plate load tests were simulated using ABAQUS 6.7 [26]. Three different traditional support models for the rigid pavement structure were considered in this study (1) type 1: a composite *k*-value model represented by a set of vertical springs with a spring coefficient k, (2) type 2: an elastic-isotropic solid layered model composed of a multiple-layered system defined by elastic modulus (*E*) and Poisson's ratio (*v*) of each layer of material, and (3) type 3: a combined model consisting of an elastic solid layer for a stabilized base layer and effective *k*-value for a compacted aggregate base and subgrade modeled by a set of vertical springs. For the FE analyses, the values obtained from the field tests at the K2 location were used as input values since K2 represented the medium range of field conditions and the *k*-value on the surface of the aggregate base at the K2 location was measured. Figure 4 shows the three types of support models and the input values used in the FE simulation. For type 1, a spring coefficient of 680 psi/in (184.6 MN/m$^3$) was selected from the static plate load test at the K2 location. For type 2, the results of FWD and DCP tests were used. For type 3, a spring coefficient of 420 psi/in (114.0 MN/m$^3$) was used, representing the effective *k*-value on top of the aggregate base layer obtained from the static plate load test at location K2 after ASB layer removal.

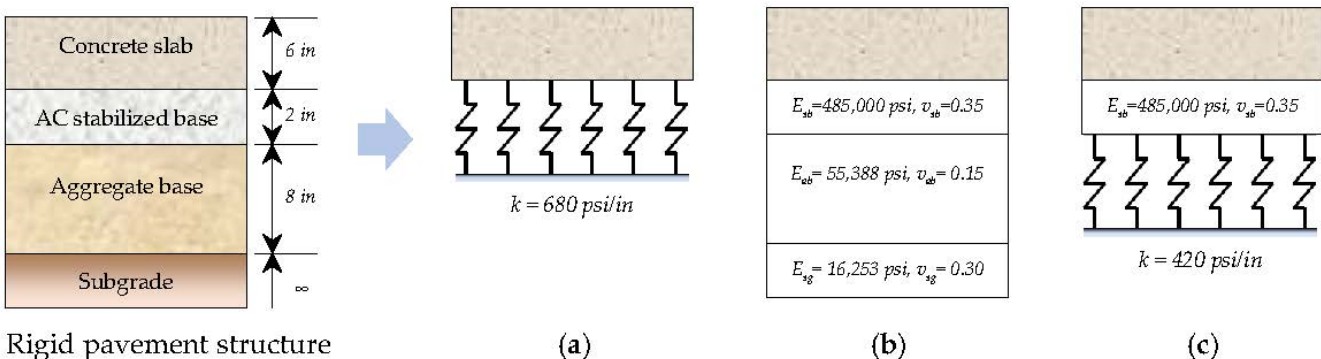

**Figure 4.** Three support models for rigid pavement system: (**a**) composite *k*-value model; (**b**) elastic-isotropic solid layered model; and (**c**) combined model (1 in = 25.4 mm, 1 psi/in = 271 kN/m$^3$).

Two different sizes of loading area (12 and 30 in (305 and 762 mm, respectively) diameter) were applied to the top surface of the 2 in (50.8 mm) the ASB layer for the three types of support models. The composite *k*-values were then computed using an applied pressure load and the corresponding average deflections at the center and edge of the loading area. Table 4 summarizes the computed values for the three support models and compares them with the field results. For the reference location K2, the field test produced a value of 1360 psi/in (369 MN/m$^3$) for the loading area 12 in (305 mm) in diameter, and a corrected *k*-value of 680 psi/in (184.6 MN/m$^3$) for the loading area 30 in (762 mm) in diameter. These field values were compared with the computed *k*-values from the FE simulations. As shown in the results, type 1 had a weakness that could not reflect the size effect of loading area. Therefore, its results are limited in interpreting actual field conditions since the composite *k*-value was affected by the size of loading area. Type 2 showed overestimated values compared with the field results for both 12 and 30 in (305 and 762 mm, respectively) loading areas. However, from type 3, the calculated values well-fitted the field data and properly reflected the size effect of loading area.

**Table 4.** Computed *k*-values on top surface of the AC stabilized layer by finite element (FE) simulation (1 in = 25.4 mm, 1 psi/in = 271 kN/m$^3$).

| Model Type | 12 in Loading Plate | | 30 in Loading Plate | |
|---|---|---|---|---|
| | *k*-Value (psi/in) | FE Simulation Field Value | *k*-Value (psi/in) | FE Simulation Field Value |
| Field test | 1360 | - | 680 | - |
| Type 1 | 680 | 50% | 680 | 100% |
| Type 2 | 3091 | 227% | 975 | 143% |
| Type 3 | 1573 | 116% | 676 | 99% |

## 4. Composition of Support Layer Properties for Determining a Desired Composite *k*-Value

To evaluate the effects of each support layer's properties on the composite *k*-value, non-repetitive static plate load tests were simulated. Based on the results of Sections 2 and 3, type 3 was used, which was the combined model consisting of an elastic solid layer for the stabilized base layer and an effective *k*-value for the compacted aggregate base and subgrade soil. In this study, three variables were considered as affecting the composite *k*-value: (1) thickness of stabilized base layer ranging from 2 to 6 in (50.8–152.4 mm); (2) elastic modulus of the stabilized base material, which was 50–2000 ksi (344–13,790 MPa); and (3) effective *k*-value of 50–300 psi/in (13.6–81.4 MN/m$^3$), which represents the characteristic of the compacted aggregate base and subgrade soil. In the FE simulations, the composite *k*-values were computed by a pressure loading of 100 psi (6.89 kPa) applied on top of the stabilized base layer and corresponding to the average deflection at the center and edge of the loading area (30 in (762 mm) diameter). The input variable ranges and the computed composite *k*-values are summarized in Table 5. The computed composite *k*-value increased as the values of the variables increased. However, as the increase rates appeared to be different depending on the variable types and values, the effect of each variable on the composite *k*-value was discussed.

**Table 5.** Input variables with values and composite *k*-values computed by the FE simulations (1 in = 25.4 mm, 1 ksi = 6.89 MPa, 1 psi/in = 271 kN/m$^3$).

| Stabilized Base Layer | | Effective *k*-Value (psi/in) | | | | | |
|---|---|---|---|---|---|---|---|
| Thickness (in) | Elastic Modulus (ksi) | 50 | 100 | 150 | 200 | 250 | 300 |
| 2 | 50 | 69 | 128 | 189 | 251 | 314 | 378 |
| | 100 | 76 | 134 | 194 | 255 | 316 | 377 |
| | 300 | 92 | 156 | 217 | 278 | 339 | 399 |
| | 500 | 103 | 172 | 236 | 299 | 361 | 422 |
| | 1000 | 124 | 201 | 272 | 340 | 406 | 470 |
| | 2000 | 154 | 244 | 324 | 400 | 472 | 542 |
| 3 | 50 | 81 | 142 | 201 | 261 | 320 | 380 |
| | 100 | 93 | 158 | 221 | 282 | 342 | 402 |
| | 300 | 124 | 202 | 274 | 341 | 407 | 472 |
| | 500 | 145 | 233 | 311 | 384 | 455 | 523 |
| | 1000 | 184 | 289 | 380 | 464 | 544 | 620 |
| | 2000 | 239 | 367 | 476 | 576 | 669 | 758 |
| 4 | 50 | 95 | 161 | 223 | 284 | 343 | 402 |
| | 100 | 114 | 188 | 256 | 320 | 384 | 445 |
| | 300 | 161 | 255 | 338 | 415 | 489 | 560 |
| | 500 | 193 | 300 | 394 | 480 | 562 | 640 |
| | 1000 | 251 | 384 | 497 | 600 | 695 | 787 |
| | 2000 | 332 | 500 | 640 | 766 | 883 | 993 |

**Table 5.** *Cont.*

| Stabilized Base Layer | | Effective *k*-Value (psi/in) | | | | | |
|---|---|---|---|---|---|---|---|
| Thickness (in) | Elastic Modulus (ksi) | 50 | 100 | 150 | 200 | 250 | 300 |
| 5 | 50 | 114 | 186 | 252 | 316 | 377 | 437 |
| | 100 | 141 | 226 | 301 | 372 | 440 | 507 |
| | 300 | 207 | 321 | 419 | 509 | 593 | 674 |
| | 500 | 250 | 382 | 494 | 596 | 691 | 781 |
| | 1000 | 330 | 498 | 637 | 762 | 878 | 986 |
| | 2000 | 441 | 660 | 837 | 994 | 1138 | 1273 |
| 6 | 50 | 133 | 214 | 286 | 353 | 418 | 480 |
| | 100 | 169 | 265 | 349 | 426 | 500 | 570 |
| | 300 | 255 | 389 | 502 | 604 | 695 | 790 |
| | 500 | 311 | 470 | 602 | 720 | 830 | 933 |
| | 1000 | 416 | 621 | 789 | 938 | 1074 | 1202 |
| | 2000 | 557 | 830 | 1049 | 1241 | 1415 | 1577 |

To identify the effects of the support layer properties on the composite *k*-value, regression analysis was performed using SPSS software [27]. In this analysis, the composite *k*-value was set as a dependent variable, and the thickness of the stabilized base layer ($T_b$, inches), elastic modulus of the stabilized base material ($E_b$, ksi), and effective *k*-value ($k_e$, psi/in) were considered as independent variables. Table 6 shows the regression coefficients for estimating the composite *k*-value. The relative effectiveness of each independent variable on the composite *k*-value cannot be directly compared using the unstandardized coefficients since the variables have different units. Thus, the standardized coefficient ($\beta$) was used, which is defined as follows:

$$\beta = Unstandardized\ coefficient \times \frac{S.D.\ of\ Dependent\ variable}{S.D.\ of\ Independent\ variable} \tag{3}$$

where *S.D.* represents standard deviation.

**Table 6.** Regression coefficients for composite *k*-value.

| Independent Variables | Unstandardized Coefficient | Standardized Coefficient |
|---|---|---|
| Constant | −395.669 | - |
| Thickness of stabilized base layer ($T_b$) | 92.335 | 0.475 |
| Elastic modulus of the stabilized base material ($E_b$) | 0.223 | 0.550 |
| Effective *k*-value ($k_e$) | 1.829 | 0.568 |

Consequently, the effective *k*-value showed the greatest effect on the composite *k*-value, followed by the elastic modulus of the stabilized base material, and then the thickness of the stabilized base layer. However, the differences among the variables were relatively very small. Therefore, it could be assumed that the effectiveness of the support layer properties on the composite *k*-value are almost similar. The regression equation can be expressed as following with a coefficient of determination ($R^2$) value of 85.1%.

$$k_\infty = 92.3T_b + 0.223E_b + 1.829k_e - 395.7 \tag{4}$$

The procedure for determining the compositions of layered support structures for satisfying a desired composite *k*-value is suggested here. In this study, the support structure was assumed to consist of a stabilized base layer and subgrade soil, with aggregate base as an option. Three types of support layer properties (i.e., $T_b$, $E_b$, and $k_e$) were considered as design variables to be determined. A conceptual procedure with the desired or targeted composite *k*-value was set first, and cases of composition satisfying the composite *k*-value were found by drawing graphs based on the results shown in Table 5. It would be im-

possible and unreasonable to finely tune an effective *k*-value and elastic modulus of the stabilized base material. In actual conditions, the effective *k*-value is generally set as a specific value after compaction according to the existing soil types. Furthermore, the elastic modulus is determined in accordance with a type of stabilized base such as concrete-treated based (CTB), ASB, lean concrete base (LCB), etc. Thus, it would be more reasonable and practical to adjust the thickness of the stabilized base layer. Figure 5 presents an example of selecting the compositions for a desired composite *k*-value. In this example, the desired composite *k*-value was set to 300 psi/in (81.4 MN/m$^3$). Based on the effective *k*-value of 50 psi/in (13.6 MN/m$^3$), three cases can be selected, satisfying the target composite *k*-value. If one stabilized base material with an elastic modulus of 500 ksi (3450 MPa) is used, a 5.8 in (147 mm) base thickness is required; another material with a 1000 ksi (6895 MPa) elastic modulus needs a 4.6 in (117 mm) thickness; another material with 2000 ksi (13,790 MPa) requires a 3.6 in (91 mm) thick stabilized base layer.

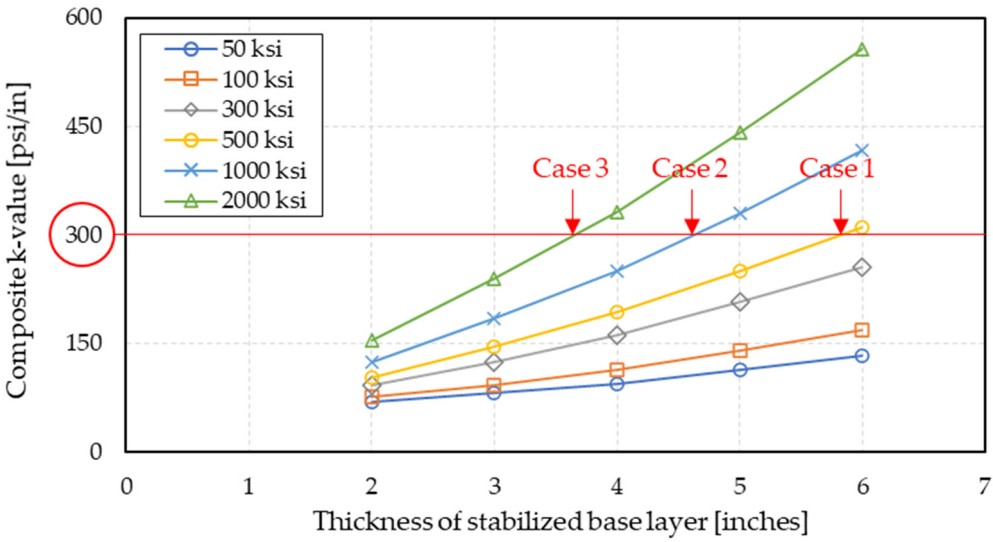

**Figure 5.** Example of selecting compositions for a desired composite *k*-value of 300 psi/in (effect of elastic modulus of the stabilized base material on the composite *k*-value when $k_e$ = 50 psi/in) (1 ksi = 6.89 MPa, 1 psi/in = 271 kN/m$^3$).

Table 7 lists the compositions of the support properties for a desired composite *k*-value: 300 psi/in (81.4 MN/m$^3$). In this example, a total of 14 cases were determined according to the effective *k*-value ranging from 50 to 200 psi/in (13.6 to 54.3 MN/m$^3$). Based on this result, if considering the cost of constructing the aggregate base and the subgrade soils having specific *k*-values (50, 100, 150, or 200 psi/in (13.6, 27.1, 40.7, and 54.3 MN/m$^3$, respectively)), the cost of the stabilized base materials having a specified elastic modulus (50, 100, 300, 500, 1000, and 2000 ksi (345, 2068, 3447, 6895, and 13,790 MPa, respectively)), and the cost for constructing a stabilized base layer with the desired thickness, the most economical composition can be selected to achieve the target composite *k*-value.

**Table 7.** Compositions for a desired composite *k*-value of 300 psi/in (1 in = 25.4 mm, 1 ksi = 6.89 MPa, 1 psi/in = 271 kN/m$^3$).

| Case No. | Effective *k*-Value (psi/in) | Stabilized Base Layer | |
| --- | --- | --- | --- |
| | | Elastic Modulus (ksi) | Thickness (inches) |
| 1 | | 500 | 5.8 |
| 2 | 50 | 1000 | 4.6 |
| 3 | | 2000 | 3.6 |
| 4 | | 300 | 4.7 |
| 5 | 100 | 500 | 4.0 |
| 6 | | 1000 | 3.1 |
| 7 | | 2000 | 2.5 |

**Table 7.** *Cont.*

| Case No. | Effective *k*-Value (psi/in) | Stabilized Base Layer | |
| --- | --- | --- | --- |
| | | Elastic Modulus (ksi) | Thickness (inches) |
| 8 | | 300 | 3.4 |
| 9 | 150 | 500 | 2.9 |
| 10 | | 1000 | 2.3 |
| 11 | | 50 | 4.5 |
| 12 | 200 | 100 | 3.5 |
| 13 | | 300 | 2.4 |
| 14 | | 500 | 2.0 |

## 5. Stress in CRCP Slab on Support System

Even though required functional and structural support characteristics for rigid pavement structures are stated, the most important factor is the level of maximum stress in concrete slab because it plays an essential role in determining the service life of the pavement system. Moreover, the level of maximum stress may depend on the various compositions of support properties even if the compositions present identical composite *k*-values. For this reason, FE analysis was additionally performed to identify the effects of various support compositions of the CRCP structure on the maximum stress induced in concrete slab under a temperature gradient and vehicle wheel loadings. In this analysis, the maximum principal stress induced in 10 CRCP slabs was considered as a criterion for the comparison.

A two-dimensional FE model of the CRCP structure was developed, as shown in Figure 6a, which was modified from previous studies [28–30]. A 4-node plane strain element was used for the CRCP slab, and longitudinal steel rebar was modeled by a 2-node beam element. To consider the effects of the stabilized base layer, the base underlying the CRCP slab was separately modeled from the subgrade using the two-dimensional 4-node elastic solid plane strain element. A 6 ft (1.83 m) long crack spacing was considered in this analysis, and the half-length of 3 ft (0.914 m) of the slab was only considered because the pavement behavior can be assumed to be symmetric with respect to the center of the slab. At the center of the slab, a vertical degree of freedom was allowed, but longitudinal and rotational displacements of the slab were restrained; for the longitudinal steel rebar, longitudinal and rotational displacements were not allowed at the ends of the steel rebar. For the stabilized base layer, the longitudinal and rotational displacements were restrained at both ends, but vertical degree of freedom was allowed. The boundary and interface conditions in the model are presented in Figure 6b–e. The subgrade layer was modeled as a set of tensionless springs to properly consider the upward curling effects (Figure 6b). For the horizontal friction resistance between the concrete slab and the stabilized base layer, and the base layer and subgrade, 145.5 psi/in (39.5 MN/m$^3$) and 22.0 psi/in (5.97 MN/m$^3$) were used, respectively (Figure 6c). To consider the behavior of the concrete slab at transverse cracks, a horizontal spring element was used (Figure 6d). It was assumed that the vertical shear and moment transfer to the adjacent concrete slab could be ignored. Horizontal movement of the concrete slab at transverse cracks was allowed within a crack width due to the expansion and contraction of the slab. An average crack width of 0.01 in (25.4 mm) was used in this analysis based on a previous field investigation [31]. A nonlinear bond slip relationship between the concrete and the longitudinal steel bar was considered in the horizontal direction (Figure 6e). An element size of 0.5 in (12.7 mm) was used in this study in accordance with a previous study of the FE modeling of CRCP, which showed good convergence in the analysis results when the element size was smaller than 1.5 in (38.1 mm) [30]. The stresses affected the CRCP slab responses were calculated at four integration points of one element, and average values were used for each element in this study. Table 8 lists the input variables and control values used in this FE analysis.

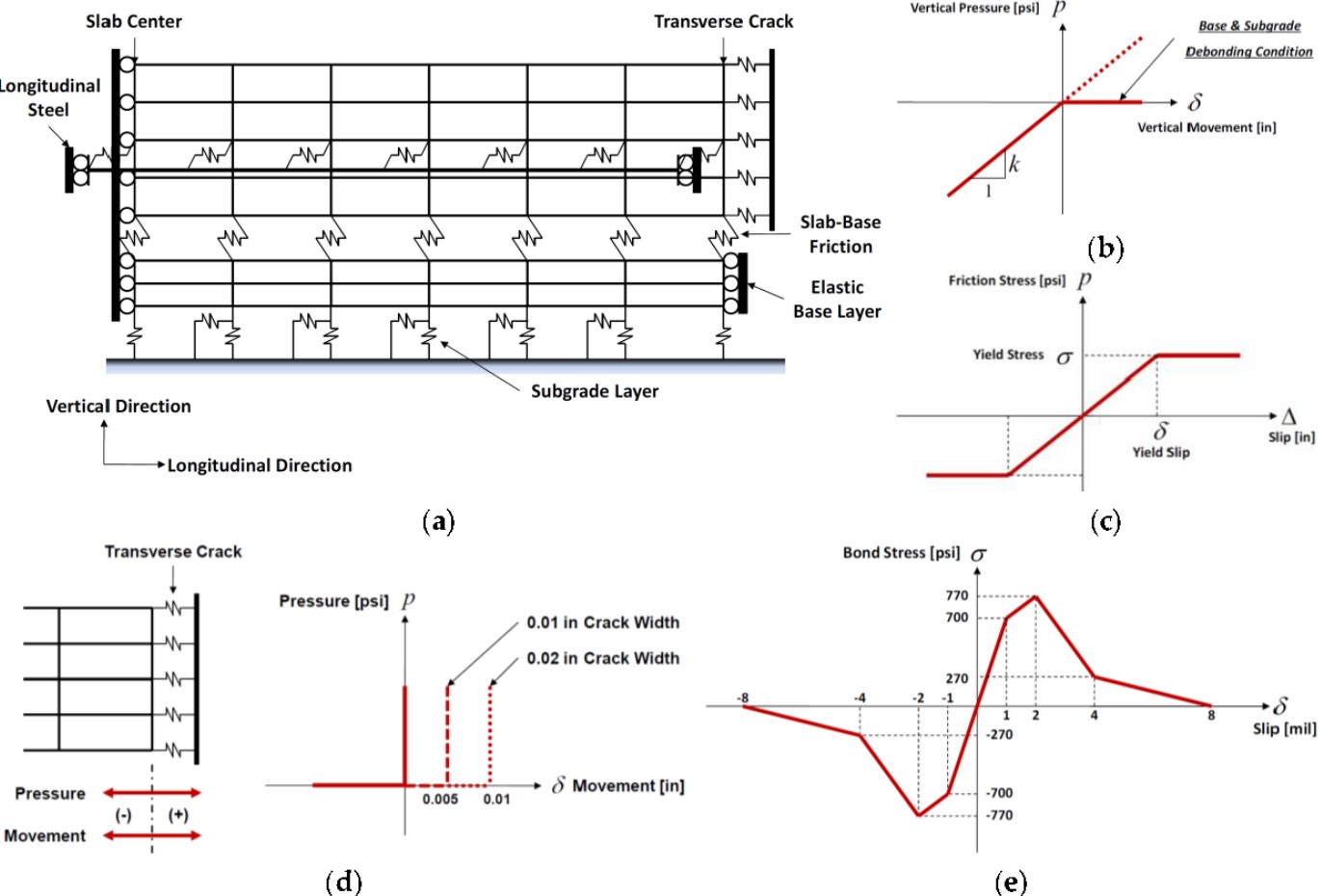

**Figure 6.** (**a**) Continuously reinforced concrete pavement (CRCP) FE model; (**b**) subgrade behavior; (**c**) friction behavior; (**d**) transverse crack behavior; and (**e**) bond-slip behavior.

**Table 8.** Input variables and control values for CRCP FE analysis.

| Variable | Value |
|---|---|
| Crack spacing | 6.0 ft (1.83 m) |
| Crack width | 0.01 in (0.254 mm) |
| Longitudinal steel spacing | 6.0 in (152.4 mm) |
| Thickness of concrete slab | 10.0 in (254 mm) |
| Depth of steel location | 5.0 in (127 mm) |
| Thickness of stabilized base | 4.0 in (101.6 mm) |
| Elastic modulus of concrete | $3.0 \times 10^6$ psi (20.7 GPa) |
| Poisson's ratio of concrete | 0.15 |
| Unit weight of concrete | 150.0 pcf (2400 kg/m$^3$) |
| CTE * of concrete | $6.0 \times 10^{-6}$/°F ($1.08 \times 10^{-5}$/°C) |
| Effective $k$-value | 150.0 psi/in (40.7 MN/m$^3$) |
| Elastic modulus of stabilized base material | $3.0 \times 10^5$ psi (2.07 GPa) |
| Poisson's ratio of stabilized base material | 0.35 |
| Unit weight of stabilized base material | 150.0 pcf (2400 kg/m$^3$) |
| CTE * of stabilized base material | $1.2 \times 10^{-5}$/°F ($2.16 \times 10^{-5}$/°C) |
| Elastic modulus of rebar | $2.9 \times 10^7$ psi (200 GPa) |
| Poisson's ratio of rebar | 0.29 |
| Unit weight of rebar | 480.0 pcf (7690 kg/m$^3$) |
| CTE * of rebar | $5.0 \times 10^{-6}$/°F ($9.0 \times 10^{-6}$/°C) |
| Diameter of rebar | 0.75 in (19.05 mm) |
| Reference temperature | 95.0 °F (35.0 °C) |

*, CTE: Coefficient of Thermal Expansion.

Both environmental and vehicle wheel loadings were considered in this analysis. Simplified temperature gradients in daytime and nighttime measured at the field test section in Cleveland, Texas, in 2004 were used as the environmental loading (Figure 7a) [31]. A typical dual-tire single-axle load with a tire pressure of 80 psi (552 kPa) was used to estimate wheel load stresses in the concrete slab. For the vehicle wheel loadings, two different loading locations were considered: center loading condition and edge (i.e., transverse crack) loading condition, as shown in Figure 7b.

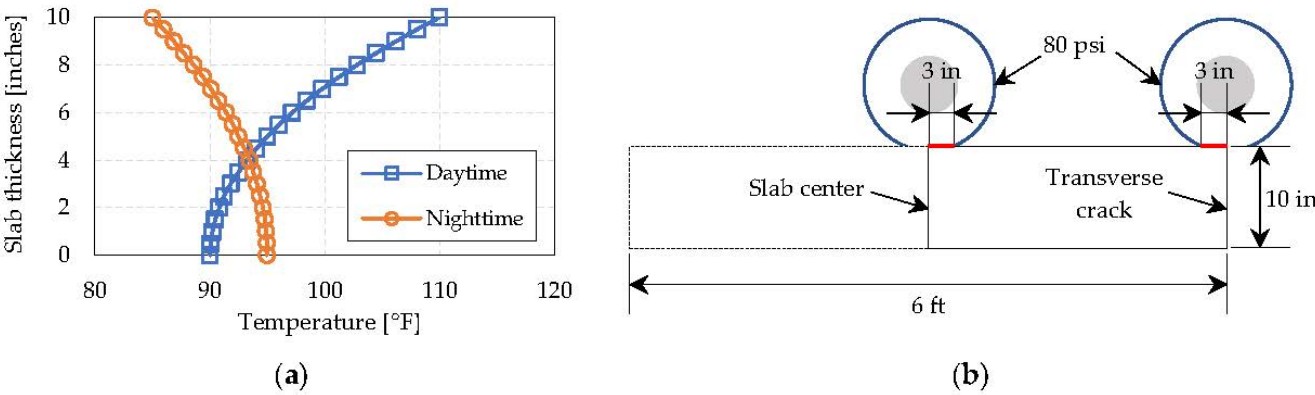

**Figure 7.** Loading conditions: (**a**) temperature gradient and (**b**) vehicle wheel loading (1 in = 25.4 mm, °C = (°F − 32)/1.8, 1 psi = 6.89 kPa).

The effects of support layer properties (i.e., $T_b$, $E_b$, and $k_e$) on the maximum stress ($\sigma_{max}$) induced in a 10 in (254 mm) thick CRCP slab were studied for both temperature gradients and vehicle wheel loadings. $T_b$ ranged from 2 to 6 inches (50.8 to 152.4 mm), $E_b$ was 50 to 2000 ksi (345 to 13,790 MPa), and $k_e$ ranged from 50 to 300 psi/in (13.6 to 81.4 MN/m$^3$). These values represent the most practical limits in the materials used in rigid pavement construction. The computed results were analyzed using regression analysis method using SPSS software, and the estimated regression coefficients are summarized in Table 9. For both temperature and vehicle wheel loadings, we found that $E_b$ had the largest effect on the maximum stress in concrete slab, followed by $T_b$, then $k_e$. The regression equations can be expressed as Equation (5) for the temperature gradient and Equation (6) for the vehicle wheel loadings with R$^2$ values of 84.0% and 82.0%, respectively.

$$\sigma_{max} = 50.6 + 4.41T_b + 0.013E_b + 0.02k_e \tag{5}$$

$$\sigma_{max} = 103.0 - 2.61T_b - 0.007E_b - 0.012k_e \tag{6}$$

**Table 9.** Regression coefficients for critical stress under temperature and vehicle wheel loadings. $T_b$, thickness of the stabilized base layer; $E_b$, elastic modulus of the base material; $k_e$, effective $k$-value.

| Independent Variables | For Temperature Loading | | For Vehicle Wheel Loading | |
|---|---|---|---|---|
| | Unstandardized Coefficient | Standardized Coefficient | Unstandardized Coefficient | Standardized Coefficient |
| Constant | 50.569 | - | 103.0 | - |
| $T_b$ | 4.407 | 0.518 | −2.607 | −0.528 |
| $E_b$ | 0.013 | 0.743 | −0.007 | −0.720 |
| $k_e$ | 0.02 | 0.141 | −0.012 | −0.150 |

Consequently, we identified that the maximum stresses induced in the CRCP slab due to temperature gradients and vehicle wheel loadings were affected by each property of the support structure. In accordance with the behavior of the CRCP slab placed on the support compositions presented in Table 7, which show that materials with identical composite $k$-values can have different properties, the maximum stresses were computed for each

composition, and are listed in Table 10. The maximum stress due to temperature gradient ranged from 71.3 to 90.7 psi (491 to 625 kPa), whereas values between 80.7 and 90.4 psi (556 and 623 kPa) were estimated for the wheel loading condition. However, the trend in the changing stress level with temperature loading were opposite that of vehicle loading. Case no. 11 produced the minimum level of the maximum stress for both temperature gradient and combined loading condition, whereas case no. 3 showed the lowest stress level for vehicle wheel loading. Thus, in this example, case no. 11 (i.e., $k_e$ = 200 psi/in (54.2 MN/m$^3$), $E_b$ = 50 ksi (345 MPa), and $T_b$ = 4.5 inches (114.3 mm)) might be considered as the optimum support composition exhibiting a composite *k*-value of 300 psi/in (81.3 MN/m$^3$). Similarly, the relative influence of slab thickness on the properties of the support structure is also significant factor for determining the optimum composition of the CRCP structure.

**Table 10.** Critical stresses due to temperature and vehicle loadings under various support property compositions with a composite *k*-value of 300 psi/in (81.3 MN/m$^3$) (1 psi = 6.89 kPa).

| | Case No. | 1 | 2 | 3 | 4 | 5 | 6 | 7 | 8 | 9 | 10 | 11 | 12 | 13 | 14 |
|---|---|---|---|---|---|---|---|---|---|---|---|---|---|---|---|
| Critical Stress (psi) | Temperature load | 86.5 | 89.1 | 90.7 | 76.5 | 77.7 | 78.6 | 80.2 | 73.4 | 74.0 | 74.7 | 71.3 | 71.6 | 72.1 | 72.2 |
| | Vehicle load | 82.4 | 81.3 | 80.7 | 87.9 | 87.4 | 87.1 | 86.5 | 89.6 | 89.3 | 89.1 | 90.4 | 90.3 | 90.2 | 90.1 |
| | Combined load | 168.9 | 170.4 | 171.4 | 164.4 | 165.1 | 165.7 | 166.7 | 163.0 | 163.3 | 163.8 | 161.7 | 161.9 | 162.3 | 162.3 |

## 6. Composition of Variables for Maximum Stress in Concrete Slab

To identify the effects of the design variables on the maximum stress in the CRCP slab, the thickness of the CRCP slab ($T_c$), the thickness of the stabilized base layer ($T_b$), the elastic modulus of the base material ($E_b$), and the effective *k*-value ($k_e$) were selected as independent variables, as shown in Table 11. All compositions of the variables totaled 900 design cases. For these various CRCP compositions, the maximum stresses were computed through FE analysis for the following loading cases: (1) the temperature gradient, (2) the vehicle wheel loading, and (3) the combined loading conditions. For loading case 3, the nighttime temperature gradient and vehicle wheel loading applied on a transverse crack produced the maximum stress level.

**Table 11.** Independent variables and values for FE analysis of the CRCP structure (1 in = 25.4 mm, 1 ksi = 6.89 MPa, 1 psi/in = 271 kN/m$^3$).

| Variable | Value |
|---|---|
| Thickness of CRCP slab, $T_c$ (in) | 6, 8, 10, 12, 14 |
| Thickness of stabilized base layer, $T_b$ (in) | 2, 3, 4, 5, 6 |
| Elastic modulus of the base material, $E_b$ (ksi) | 50, 100, 300, 500, 1000, 2000 |
| Effective k-value, $k_e$ (psi/in) | 50, 100, 150, 200, 250, 300 |

Using the sets of computed maximum stresses induced in the CRCP slab for the three loading cases, regression analyses were conducted, and the relative effects of each property on the maximum stress generations were discussed. Table 12 shows the estimated regression coefficients for all loading cases. According to the standardized coefficients, the absolute level of relative influence of each variable on the maximum stress for all loading cases are in the following order: $T_c > E_b > T_b > k_e$. We found that the thickness of the concrete slab tended to govern the maximum stress more significantly under the combined loading condition than the other loading conditions. The stress analysis results produced the regression Equation (7) for the temperature gradient, Equation (8) for the vehicle wheel loading, and Equation (9) for the combined loading with R$^2$ values of 88.3%, 88.3%, and 92.4%, respectively.

$$\sigma_{cr} = 117.5 - 6.26T_c + 4.39T_b + 0.013E_b + 0.024k_e \tag{7}$$

$$\sigma_{cr} = 308.8 - 17.61T_c - 4.54T_b - 0.013E_b - 0.027k_e \tag{8}$$

$$\sigma_{cr} = 391.3 - 20.25T_c - 1.337T_b - 0.003E_b - 0.014k_e \tag{9}$$

**Table 12.** Regression coefficients for the maximum stress under temperature, vehicle wheel, and combined loading conditions.

| Independent Variables | Un-Standardized Coefficient | | | Standardized Coefficient | | |
|---|---|---|---|---|---|---|
| | **Temperature** | **Vehicle** | **Combined** | **Temperature** | **Vehicle** | **Combined** |
| Constant | 117.504 | 308.834 | 391.301 | - | - | - |
| $T_c$ | −6.256 | −17.614 | −20.25 | −0.798 | −0.918 | −0.960 |
| $T_b$ | 4.391 | −4.535 | −1.337 | 0.280 | −0.118 | −0.032 |
| $E_b$ | 0.013 | −0.013 | −0.003 | 0.401 | −0.157 | −0.035 |
| $k_e$ | 0.024 | −0.027 | −0.014 | 0.091 | −0.043 | −0.020 |

## 7. Optimized Layer Composition of CRCP

For a functional purpose such as preventing the erosion of support materials and supplying adequate construction platform, it is highly recommended to construct a stabilized support structure for rigid pavement systems. In this regard, diverse chemical treatments for subgrade soils and construction of stabilized base layer have been widely used in the pavement construction industry, and the composite *k*-value of the support system is used for ensuring the achievement. For this reason, a desired composite *k*-value should be confirmed first, and the compositions of the support layer properties satisfying the composite *k*-value should be subsequently determined. Afterward, an allowable maximum stress is determined that can satisfy mixed traffic and environmental loading conditions during a designated service life. For all compositions of the support structure previously selected, a minimum thickness of CRCP slab is determined that can produce the maximum stress not exceeding the allowable stress. Among the cases, the most economical composition is selected. Figure 8 illustrates the schematic procedure to determine the optimized layer composition of the CRCP structure.

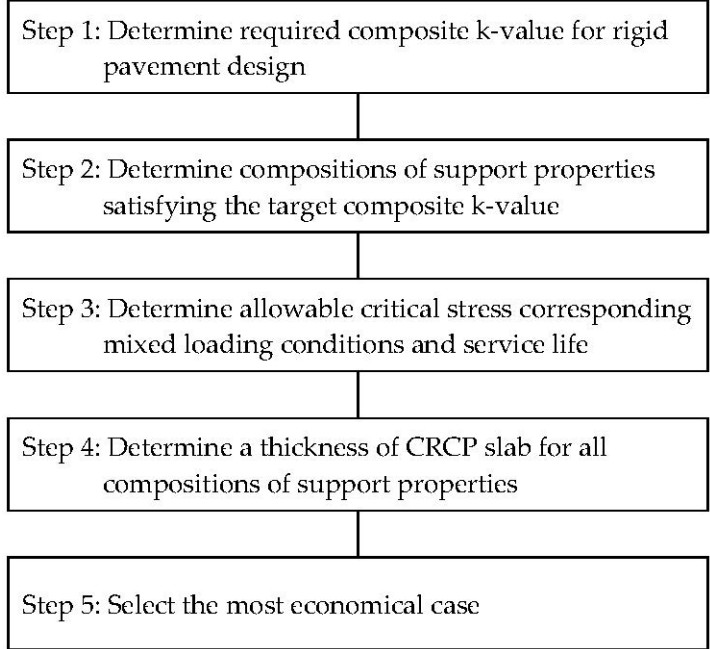

**Figure 8.** Procedure to determine optimized composition of CRCP structure.

A case study was conducted as follows: First, a composite *k*-value of 300 psi/in (81.3 MN/m$^3$) was selected, and the compositions of the support layer properties (thickness of stabilized base layer, elastic modulus of the base material, and effective *k*-value)

producing the composite $k$-value of 300 psi/in (81.3 MN/m³) were determined, as shown in Table 7. To determine allowable maximum stress, a number of load applications ($N$) that represents the traffic volume for the pavement design should be considered. Most damage occurs as fatigue damage in rigid pavement systems. The general expression for fatigue damage accumulation is as follows

$$FD = \sum \frac{n}{N}, \tag{10}$$

where $FD$ = total fatigue damage, $n$ = applied number of load applications, and $N$ = allowable number of load applications.

The applied number of load applications is the actual number of passed traffic load, and the allowable number of load applications is the number of load cycles at which fatigue failure is expected. The allowable number of load applications can be determined using the following fatigue model [19]

$$\log(N) = C_1 \cdot \left(\frac{MR}{\sigma}\right)^{C_2} + 0.4371, \tag{11}$$

where $N$ = allowable number of load applications, $MR$ = concrete modulus of rupture, $\sigma$ = applied stress, $C_1$ = calibration constant = 2.0, and $C_2$ = calibration constant = 1.22.

The Texas Pavement Manual has regulated that an $MR$ of 620 psi (4.27 MPa) at 28 days should be used for concrete pavement design [2]. In this example, three different values of the number of load applications were considered. The allowable maximum stresses ($\sigma_{\text{allow}}$) corresponding to the three cases of load applications were calculated using Equation (10), and the stress/strength ratio for the three cases is presented in Table 13.

**Table 13.** Allowable stresses and stress/strength ratios corresponding each number of load applications (1 psi = 6.89 kPa).

| Case No. | Number of Load Applications | Allowable Stress (psi) | Stress/Strength Ratio |
|---|---|---|---|
| 1 | 2,500,000 | 253.3 | 0.41 |
| 2 | 250,000 | 294.4 | 0.47 |
| 3 | 25,000 | 354.1 | 0.57 |

To find the acceptable minimum thickness of a concrete slab that produces a maximum stress that is less than the computed allowable stress for the three load application cases, the maximum principal stresses were calculated by changing the slab thickness under the combined loading condition for the selected 14 compositions of support layer properties. Table 14 presents the layer compositions of the CRCP structure that do not exceed the allowable stress for the combined loading condition and the different load application cases. Considering construction feasibility, the CRCP slab thickness is expressed as an integer number.

An optimized composition of CRCP structure should consider the financial aspect. Accordingly, the most economical composition resulting in the lowest initial construction cost among the 14 cases was considered the optimized layer composition of the CRCP structure for each different traffic volume. Based on the average low-bid unit prices from the Texas Department of Transportation [32], the initial CRCP construction costs for some selected cases were calculated, as shown in Figure 9. Standardized stabilized base materials were used: asphalt stabilized base (ASB), cement-treated base (CTB), lime-treated base (LTB), and lean concrete base (LCB). Table 15 summarizes the selected optimized layer compositions of the CRCP structure with initial construction costs for the three design traffic volumes. This case study shows the schematic procedure for selecting the optimized layer composition for a CRCP structure with consideration of cost. Under field conditions,

the compositions might be adjusted based on the constructability of the material and the thickness of each layer.

**Table 14.** Layer compositions of CRCP structures under the combined loading condition (1 in = 25.4 mm, 1 ksi = 6.89 MPa, 1 psi/in = 271 kN/m$^3$).

| Case No. | Effective *k*-Value (psi/in) | Stabilized Base Layer | | Thickness of Concrete Slab (inches) Number of Load Applications | | |
|---|---|---|---|---|---|---|
| | | Elastic Modulus (ksi) | Thickness (inches) | 2,500,000 | 250,000 | 25,000 |
| 1 | | 500 | 5.8 | 9 | 7 | 7 |
| 2 | 50 | 1000 | 4.6 | 9 | 7 | 7 |
| 3 | | 2000 | 3.6 | 9 | 7 | 7 |
| 4 | | 300 | 4.7 | 8 | 7 | 6 |
| 5 | | 500 | 4.0 | 9 | 7 | 6 |
| 6 | 100 | 1000 | 3.1 | 9 | 7 | 6 |
| 7 | | 2000 | 2.5 | 9 | 7 | 6 |
| 8 | | 300 | 3.4 | 8 | 7 | 6 |
| 9 | 150 | 500 | 2.9 | 8 | 7 | 6 |
| 10 | | 1000 | 2.3 | 8 | 7 | 6 |
| 11 | | 50 | 4.5 | 8 | 7 | 6 |
| 12 | 200 | 100 | 3.5 | 8 | 7 | 6 |
| 13 | | 300 | 2.4 | 8 | 7 | 6 |
| 14 | | 500 | 2.0 | 8 | 7 | 6 |

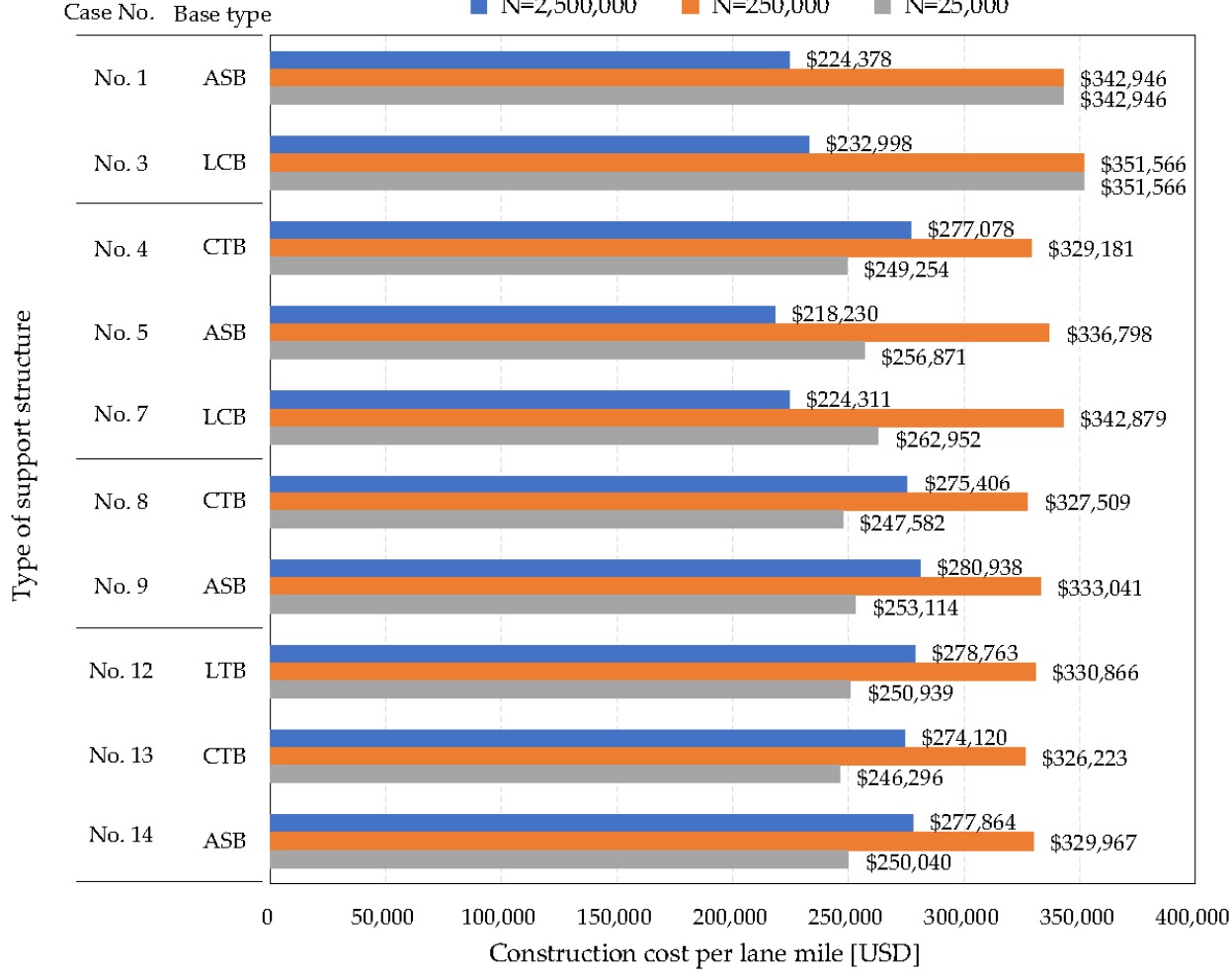

**Figure 9.** CRCP initial construction cost per lane mile.

**Table 15.** Optimized layer compositions and initial construction costs of a CRCP structure for three cases of traffic volumes (1 in = 25.4 mm, 1 psi/in = 271 kN/m$^3$).

| Case No. | 1 | 2 | 3 |
|---|---|---|---|
| Effective $k$-value (psi/in) | 100 | 200 | 200 |
| Type of stabilized base layer | ASB * | CTB ** | CTB ** |
| Thickness of stabilized base layer (in) | 4 | 2.4 | 2.4 |
| Thickness of concrete slab (in) | 9 | 7 | 6 |
| Initial construction cost (USD/lane mile) | 218,230 | 326,223 | 246,296 |

*, ASB: asphalt stabilized base; **, CTB: cement treated base.

## 8. Conclusions

In this paper, a procedure for determining the optimized layer composition of CRCP was proposed that considers the structural characteristics of the support system and the cost aspect. To achieve the research purpose, field tests, FE analysis, and regression analysis were conducted, and the results are summarized below:

- The FWD test showed non-uniform stiffness distribution on top of the ASB layer. The DCP data showed the boundary of the aggregate base and subgrade layer. Whereas the aggregate base was stronger than subgrade, the subgrade layer exhibited a more uniform condition than the aggregate base. The composite $k$-values on top of ASB layer ranged from 680 to 920 psi/in (184.3 to 249 MN/m$^3$), although the $k$-value on top of the aggregate base was 420 psi/in (113.8 MN/m$^3$). The 2 in (50.8 mm) asphalt stabilized base improved the stiffness of the support system about 1.5 times in this field test.
- Through FE analysis, we identified that the effective $k$-value had the largest effect on the composite $k$-value, followed by the elastic modulus of the stabilized base material, and then the thickness of the stabilized base layer. A regression equation was derived for estimating the composite $k$-value. In addition, a procedure to determine the compositions of the support layer properties for a desired composite $k$-value was proposed.
- The FE analysis results showed that even though the support systems had identical composite $k$-value, the maximum stress induced in the CRCP slab varied according to the compositions of the support layer properties under both temperature gradient and vehicle wheel loading. As the CRCP slab thickness increased, the maximum stress in the concrete slab decreased, where the stresses due to vehicle wheel loading decreased more significantly than those due to the temperature gradient.
- The regression relationships were derived to estimate the maximum stress in the CRCP slab for temperature gradient, vehicle wheel loading, and combined loading conditions. For all loading cases, the effect of concrete slab thickness was overwhelmingly larger than those of the other properties, including stabilized base thickness, elastic modulus of the base material, and effective $k$-value. Based on these results, a procedure for selecting an optimized layer composition of CRCP structure was suggested with consideration of the economical aspect.

**Author Contributions:** Conceptualization, B.H.C.; methodology, B.H.C. and M.W.; formal analysis, B.H.C.; investigation, B.H.C.; resources, B.H.C. and M.W.; data curation, B.H.C.; writing—original draft preparation, B.H.C.; writing—review and editing, B.H.N.; visualization, B.H.C.; supervision, M.W. and B.H.N. All authors have read and agreed to the published version of the manuscript.

**Funding:** This research received no external funding.

**Institutional Review Board Statement:** Not applicable.

**Informed Consent Statement:** Not applicable.

**Data Availability Statement:** The data used are available upon request.

**Conflicts of Interest:** The authors declare no conflict of interest.

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
