# Peer review of "Layer Composition of Continuously Reinforced Concrete Pavement Optimized Using a Regression Analysis Method"

_infrastructures, doi:10.3390/infrastructures6040056_

Round 1
Reviewer 1 Report
The paper presents a comprehensive research about continuously reinforced concrete pavements, with particular reference to the support/subgrade. Based on an in-situ survey campaign consisting in various structural tests (FWD, DCP, static plate), various finite element models were set up to describe the composite k-value. Then, k-value of support system was linked to the thickness of the base layer, its elastic modulus and effective k-value through regression analysis methods. Furthermore, the stress in the concrete surface slab was evaluated as a function of its properties and the characteristics of the supports. Finally, a cost analysis was employed to propose a method for the selection of suitable and cost-containing pavement structures.
After the review, neither significant comments are addressed to the conceptual contents of the manuscript, nor improvements of paragraphs are required. Small adjustments and additions are suggested, together with text/figure corrections (see the following list):
- Small text errors have been found (e.g., Line 12 “were”; Line 69 “case”; Line 87 “an”; Line 123 “presents”; etc.). Authors are invited to perform a quick text check to fix all these aspects.
- Figure 3, Line 179: it is quite obvious that “AC” acronym is referred to “ASB” (Asphalt Stabilized Base). But, I think that it can be directly substituted everywhere with “ABS” for the sake of uniformity.
- Figure 4, Figure 7b: there are some cut words. Maybe, the fact could be given by the pdf conversion. However, Authors could check the problem in their source images for safety.
- Figure 5: in the present version of pdf, the red indications (circle for 300 psi/in, horizontal line, vertical arrows, and case labels) are clearly out of their correct position (clearly, a left shift is occurred). Maybe, the fact could be given by the pdf conversion. However, Authors should check this aspect.
- Figure 6: given the number of sub-figures – (a) to (e) – and their positions, the division of the Figure in two distinct pages could make difficult data reading. Maybe, authors can change the Figure position (this could be optional).
- Paragraph 7: talking about the cost analysis, a list in the text of some unitary costs of the considered materials could be given. Author could insert it for the sake of completeness.
- Figure 9: label of left y-axis is overlayed with numbers. Authors should fix this aspect.
Line 515: Reference number 24 is the same of number 16. Authors must fix this aspect, rearranging the reference’s numeration.
Author Response
The authors appreciate your efforts in reviewing this paper and thoughtful comments. All the comments were answered by authors’ best. The manuscript was appropriately edited to improve content clarity and readability.
Please find the attached file, the authors' response.

Reviewer 2 Report
Dear Authors, please see in the attached file.

Author Response

(The authors gave the same response as above.)

Reviewer 3 Report
The paper entitled ”Optimized Layer Composition of Continuously Reinforced Concrete Pavement Using Regression Analysis Method” deals with an interesting subject as it presents a research based on a statistical method for more conclusive and economically efficient results.
The Abstract is well structured and it includes on short the aim of the paper, the study methods and its results.
Introduction presents the general notions, in order to make a general view on the subject and to understand its importance. The Introduction concludes with the aim of the paper.
In Field Investigation chapter, there are described in detail the methods applied in conducting the study and the results obtained. In Evaluation of Support Models chapter, the results are statistically analyzed, and in 4th chapter, Composition of Support Layer Properties for Determining a Desired Composite k- 197 value, are determined optimum compositions for CRCP that are further analyzed in 5th chapter.
The Conclusions highlight the results obtained.
Author Response

(The authors gave the same response as above.)
